# Temporal and Spatial Variation Characteristics of the Fish Biomass Particle-Size Spectra in the Shandong Province Area of the Yellow River

**DOI:** 10.3390/biology14020196

**Published:** 2025-02-13

**Authors:** Lufeng Sun, Jianglong Que, Jianqun Niu, Xiuqi Li, Junpeng Wang, Xuri Cong

**Affiliations:** 1Shandong Freshwater Fisheries Research Institute, Jinan 250013, China; bensun236@163.com (L.S.);; 2Aquatic Conservation and Rescue Center of Jiangxi Province, Nanchang 330029, China; 3Institute for the Control of Agrochemicals of Shandong Province, Jinan 250131, China; 13589116346@139.com

**Keywords:** fish, biomass particle-size spectrum, particle-size structure, ABC curve, Yellow River

## Abstract

This study examined changes in fish communities in the Shandong section of the Yellow River from summer 2022 to spring 2023. Small fish dominated in the spring and summer, medium fish in autumn, and large fish in winter. Upstream areas had more small fish, while the estuary, with fewer disturbances, supported larger species. Human activities, especially overfishing, caused severe stress on fish populations, shifting them toward smaller, fast-growing species and reducing stability. This research highlights the need to protect fish populations and offers insights for the sustainable management of Yellow River fish resources, benefiting biodiversity, food security, and local livelihoods.

## 1. Introduction

The concept of particle-size spectra was first proposed by Sheldon and Parsons [1], representing the curve that illustrates the relationship between biomass or abundance and the species’ particle size [2]. This curve reflects the characteristics of biomass, abundance, and particle-size structure through its fluctuations, making it an important method in aquatic ecosystem research [3,4]. Since the introduction of particle-size spectra theory into the study of aquatic ecosystems, fish particle-size spectra have been widely applied to assess the structural characteristics of fish communities across different trophic levels and the impacts of human disturbances on ecosystems [2]. Much research has been conducted internationally in this area. For instance, Duplisea and Kerr used fish particle-size spectrum parameters to explain how natural disturbances affect the structure of deep-sea fish communities [5]. Graham and others studied the effects of fishing on coral reef fish communities through fish particle-size spectra, finding a negative correlation between fish abundance and fishing intensity within specific size classes [6]. Jung and Houde’s research on fish communities in the Chesapeake Bay revealed a “bimodal” fish biomass particle-size spectrum, with the first peak corresponding to small zooplanktivorous fish and the second peak representing larger omnivorous fish [7]. Blanchard and colleagues established particle-size spectrum models based on four fish populations in the North Sea to evaluate community responses to fishing and to conduct environmental monitoring [8]. In China, extensive research has been conducted on the particle-size spectra of phytoplankton [9,10] and marine benthic organisms [11,12], dating back several years, while studies on fish biomass particle-size spectra have only recently gained attention [2,3,13].

The Yellow River, known as one of the world’s most sediment-laden rivers, has relatively limited fish resources compared to the Yangtze and Pearl River systems [14,15,16]. Moreover, research on fish resources in the Yellow River remains scarce, with existing studies being sporadic and covering only certain segments of the main stream or localized areas. These efforts are insufficient to comprehensively reflect the overall status of fish communities in the Yellow River. This study adopted the fish biomass particle-size spectrum approach due to its significant advantages over traditional fishery resource assessment methods. Traditional methods typically focus on individual species or specific groups, making it difficult to comprehensively reveal the overall structure and dynamic functions of the ecosystem. In contrast, the particle-size spectrum method integrates biomass and abundance across different size classes, enabling it to more sensitively detect changes in community structure and assess the potential impacts of fishing activities and environmental disturbances on ecosystems. Particularly in data-limited waters, the biomass particle-size spectrum can circumvent the reliance on traditional methods for age determination and species-specific growth parameters [11].

Against this backdrop, this study, based on fish samples collected from the summer and autumn of 2022 to the winter and spring of 2023, constructed a fish biomass particle-size spectra using the particle-size spectrum method. It analyzed the characteristics of the fish particle-size structure in the Shandong section of the Yellow River and compared the seasonal variations in the particle-size structure of fish communities. By revealing the current state of fish diversity in this region, this study not only provide essential baseline data and scientific evidence for the sustainable development and effective management of fishery resources in the Shandong section of the Yellow River but also demonstrates the potential and advantages of the fish biomass particle-size spectrum approach in studying complex aquatic ecosystems.

## 2. Survey and Research Methods

### 2.1. Survey Time and Method

According to the *Fishing Resource Survey Standards for the Yellow River Basin*, this study established four survey sections in the lower reaches of the Yellow River, including the protected reach (Gaocun), the Dongping Lake reach (Dong’a), the curved reach (Gaoqing), and the estuarine section (Figure 1). These sections were set up to comprehensively cover the study area and reflect the overall distribution of fish communities. To reveal the seasonal variations in fish communities, continuous fishery resource surveys were conducted from the summer and autumn of 2022 to the winter and spring of 2023 across these four sections. The sampling was independently conducted by the research team, and to ensure the comprehensiveness and representativeness of the data, a combination of bottom and surface gill nets was used for random sampling. The nets were 50 m in length, 2 m in height, with mesh sizes of 1 cm, 2 cm, and 4 cm. Considering the high sediment content of the Yellow River, which could affect the efficiency of fishing, the net deployment time was strictly limited to 4 h per day. Species identification and classification followed the standards established by Cheng Qingtai, Zheng Baoshan [17], and Nelson [18], providing scientifically accurate data support and a reliable basis for classification.

### 2.2. Research Methods

#### 2.2.1. Abundance–Biomass Comparison (ABC) Curve

The ABC curve method was first proposed by Warwick, and analyzes the status of biological communities under different levels of human disturbance by comparing the trends of biomass and abundance dominance curves on the same coordinate system [19]. The communities are categorized into three states: undisturbed, moderately disturbed, and severely disturbed [20]. When the biomass dominance curve is above the abundance dominance curve, the community is in an undisturbed state, primarily consisting of slow-growing, late-maturing large species. When the biomass and abundance dominance curves intersect, the community is in a moderately disturbed state. Conversely, when the biomass dominance curve is below the abundance dominance curve, the community is in a severely disturbed state, predominantly featuring fast-growing, early-maturing small species [21,22].

The total data of all species from the sampling sites were used as samples to calculate the ABC curve for the surveyed water area, with *W* representing the statistical measure of the relative relationship between biomass and abundance [23]:$$\begin{array}{l} W=\sum_{i=1}^{S}\frac{\left(Bi-Ai\right)}{50(S-1)} \end{array}$$
where *A_i_* and *B_i_* are the cumulative percentages of abundance and biomass, respectively, corresponding to species rank *i*, and *S* is the number of species in the survey season.

#### 2.2.2. Biomass Particle-Size Spectra

The biomass particle-size spectrum mainly consists of two types: the Sheldon-type biomass particle-size spectrum and the normalized biomass size spectrum (NBSS). Both types use the upper limit values of particle sizes transformed by log_2_ on the horizontal axis, while there are differences in the vertical axes. The Sheldon-type spectrum uses the total biomass corresponding to the unit fishing effort (net·day) transformed by log_2_ as the vertical coordinate [21], while the normalized type uses the ratio of biomass to the particle-size width corresponding to the unit fishing effort (net·day) as the vertical coordinate [23].

The Sheldon-type spectrum reflects the particle-size structure characteristics of fish communities based on the “peak shape” formed by the peaks and troughs on the curve [2]. In an ideal scenario, the points across different particle-size classes in the normalized type exhibit a linear trend, with a slope of −1 [17]. However, when the community is disturbed by external factors, the spectrum line takes on a “dome” parabolic shape [5,23,24], with the curvature influenced by factors such as the productivity level of the surveyed waters [25], habitat conditions [23], particle-size classes [26], and fishing intensity [27].

#### 2.2.3. Resource Density

The resource density *D* (g/net/day) at each site was estimated using the unit fishing effort catch method [28]:*D = C/n*
where *C* represents the actual catch amount for each site over 4 h of net fishing (g·(net·day)^−1^), and *n* is the number of nets at each sampling site.

### 2.3. Data Processing

Before data analysis, the raw data underwent standardization, converting the catch weight and number of fish at each site for each season into daily catch values per unit fishing effort. The classification of fish particle-size classes was based on the method proposed by Sheldon et al., which uses a geometric series with a common ratio of 2, categorizing the fish species into different particle-size classes (unit: g) based on their body weight [29]. The biomass particle-size spectrum was constructed by converting the biomass units calculated from unit fishing effort into g/net/day for subsequent data analysis and plotting. Dominant species were defined using the Index of Relative Importance (*IRI*) proposed by Pinkas et al., with an IRI greater than 1000 indicating significant dominance [30]. The average body weight of fish in each season was calculated by dividing the total catch weight for the season by the number of fish caught. The significance of differences in species composition, ABC curves, and biomass particle-size spectra across different years was tested using one-way Analysis of Similarities (ANOSIM) in PRIMER 5.0 software and one-way Analysis of Variance (ANOVA) in SPSS 19.0 software [31].

## 3. Results

### 3.1. Fish Species Composition

The number of fish species caught in the summer of 2022, autumn of 2022, winter of 2023, and the spring of 2023 were 34, 32, 23, and 36 species, respectively, all belonging to the class Actinopterygii. The species richness at the order, family, and genus levels in the summer of 2022 was higher than in the other three seasons. Among the four surveys, the Cypriniformes order was the most prevalent, with 20, 20, 18, and 23 species, respectively, accounting for 58.82%, 62.50%, 78.26%, and 63.89% of the total fish species in each season.

In the summer of 2022, the catch was mainly composed of small-sized fish such as *Coilia nasus*, *Hemibarbus maculatus*, *Parabramis pekinensis*, and *Saurogobio dabryi*. In autumn, the dominant species included *Pseudobrama simoni*, *Carassius auratus*, *Parabramis pekinensis*, *Hemiculter bleeker*, and *Squaliobarbus curriculus*. Winter catches were dominated by larger-sized fish such as *Cyprinus carpio*, *Hypophthalmichthys molitrix*, *Carassius auratus*, and *Squaliobarbus curriculus*. In spring, the dominant species included *Pelteobagrus nitidus*, *Pseudobrama simoni*, *Carassius auratus*, *Coilia nasus*, and *Parabramis pekinensis* (Table 1). The replacement of dominant species throughout the year was significant (*p* < 0.05).

The catch per unit effort (CPUE) was highest in autumn 2022 and lowest in winter, while the catch weight was highest in winter and lowest in autumn.

### 3.2. ABC Curve

The differences in the ABC curves for the fish communities in the Shandong province area of the Yellow River during the summer of 2022, autumn of 2022, winter of 2023, and spring of 2023 were not significant. The biomass dominance curves were all located below the abundance dominance curves, indicating that the fish communities in the Shandong province area of the Yellow River were in severely disturbed states during all four seasons. The relative relationship between biomass and abundance, measured by the W value, was less than 0 for all seasons, with values of −0.16, −0.13, −0.19, and −0.11, respectively (Figure 2).

### 3.3. Seasonal Variation of the Fish Biomass Particle-Size Spectra

Based on the survey results from the four seasons from 2022 to 2023, the seasonal variation of the fish biomass particle-size spectrum in the Shandong province area of the Yellow River was constructed. The Sheldon-type biomass particle-size spectra was generally unimodal, while the normalized biomass particle-size spectra fitting curves exhibited a dome-shaped parabolic pattern (Figure 3).

In spring and summer, the range of fish particle sizes was consistent and higher than in autumn and winter. There were significant differences in the highest peak values of the Sheldon-type biomass particle-size spectra among the four seasons: both spring and summer peaks were between 32–64 g, the autumn peak was 64–128 g, and the winter peak was 1024–2056 g. The fitting curves of the standardized biomass particle-size spectra showed differences, with an overall arrangement of the biomass particle-size spectra lines from high to low in the order of spring, summer, autumn, and winter (Figure 3). The curvature was largest in autumn and smallest in winter, while the coefficient of determination *R*^2^ was highest in spring and lowest in summer (Table 2).

From 2022 to 2023, the majority of the particle-size classes exhibited a gradual decreasing trend in individual numbers from spring to winter, and significant interannual variations in the main species composition within each particle-size class were observed. For example, in the particle-size class of 32–64 g, the summer of 2022 was primarily dominated by *Coilia nasus* and *Hemibarbus maculatus*. In autumn, the dominant species shifted to *Pseudobrama simoni*, *Carassius auratus,* and *Squaliobarbus curriculus*. In winter, *Carassius auratus* and *Squaliobarbus curriculus* predominated, while in spring, the dominant species included *Pelteobagrus nitidus*, *Pseudobrama simoni*, *Carassius auratus,* and *Coilia nasus*.

### 3.4. Spatial Distribution Characteristics of Fish Biomass Particle-Size Spectra

(1) Sheldon-type Fish Biomass Particle-size Spectra: The annual Sheldon-type fish biomass particle-size spectra in the Shandong province area of the Yellow River exhibited irregular sawtooth shapes, generally resembling a unimodal pattern (Figure 4). There were differences in the parameters of the Sheldon-type fish biomass particle-size spectra among the different survey areas. In Gaocun and Gaoqing, the fish particle-size classes ranged from 0–1024 g, with both areas showing peak particle sizes corresponding to the 16–32 g class, dominated by *Pseudobrama simoni* and *Carassius auratus.* In Dong’a, the fish particle-size classes ranged from 0–2048 g, with peak sizes also corresponding to the 16–32 g class, primarily featuring *Coilia nasus* and *Carassius auratus.* The estuarine fish particle-size range was the widest, from 0–4096 g, with peak sizes corresponding to the 512–1024 g class, and consisting predominantly of *Cyprinus carpio*, *Hypophthalmichthys molitrix*, and *Liza haematocheila* (Table 3).

(2) Normalized Fish Biomass Particle-size Spectra: The differences in the standardized fish biomass particle-size spectra across the various survey areas in the Shandong province of the Yellow River were not significant. The annual fitting curves for each area exhibited a dome-shaped parabolic pattern. The curvature values for the four survey areas ranged from −0.18 to −0.11, with the coefficients of determination *R*^2^ between 0.92 and 0.94. Among these, the estuarine waters had the highest curvature, followed by Dong’a, while Gaoqing exhibited the lowest. In the smaller particle-size range of 0–32 g, biomass was highest in Dong’a and lowest in the estuarine waters. Conversely, in the larger particle-size range of 256–4096 g, biomass was highest in the estuarine waters and lowest in Gaoqing (Table 3).

## 4. Discussion

### 4.1. Status of the Fish in the Shandong Province Area of the Yellow River

The results of this study’s ABC curve indicated that the fish community in the Shandong province area of the Yellow River was in a severely disturbed state from 2022 to 2023. The species composition was primarily dominated by fast-growing and early-maturing low-value fish such as *Pseudobrama simoni* and *Hemibarbus maculatus*. The relative relationship between biomass and abundance, measured by the W value, was highest in spring, followed by autumn, and lowest in winter. This variation was primarily related to the fish population’s own replenishment mechanisms, growth habits, changes in climate, and human activities.

Previous studies have suggested that in spring, species such as *Pseudobrama simoni*, *Hemibarbus maculatus*, *Carassius auratus*, *Squaliobarbus curriculus*, *Coilia nasus,* and *Parabramis pekinensis* spawn in large quantities. Since the implementation of ecological water scheduling in the Yellow River, the spawning volume and juvenile fish population in the Shandong province have significantly increased, a finding further corroborated by this research.

Additionally, since the introduction of a fishing ban system, fishing prohibition has effectively protected juvenile fish in the waters, allowing them to grow and fatten. For example, in the summer of 2022, the percentage of *Pseudobrama simoni* and *Carassius auratus* in the species composition ranked first and second, with respective percentages of 16.12% and 13.18%. The fishing ban system has, to some extent, facilitated the recovery of fish resources, resulting in higher catch volumes of smaller-sized species within the fish community composition of the Shandong province area of the Yellow River. The dominant species consisted primarily of smaller-sized fish. In contrast, due to the higher economic value of larger fish, fishermen along the Yellow River often selectively target larger individuals. This has resulted in significantly greater fishing pressure on larger fish compared to smaller species. Consequently, larger species such as *Cyprinus carpio*, *Ctenopharyngodon idella*, *Hypophthalmichthys molitrix*, *Aristichthys nobilis*, and *Liza haematocheila* have experienced resource depletion, leading to significant seasonal variations in their catch volumes [32].

As fishing pressure increases, the species composition of fish communities has shifted from slow-growing, late-maturing, and larger-sized species to fast-growing, early-maturing, and smaller-sized species [33,34]. The catch of larger individual fish has decreased, while the catch of smaller individual fish has increased, resulting in changes in the structure of the fish community [35].

### 4.2. Characteristics of the Fish Biomass Particle-Size Spectra in the Shandong Province Area of the Yellow River

The Sheldon-type fish biomass particle-size spectra can reflect changes in fish species composition, biomass, and abundance across different particle-size classes [21]. From 2022 to 2023, there were significant seasonal shifts and spatial differences in fish community composition across various particle-size classes. In spring, the dominant fish species were primarily concentrated in the 32–64 g size class, mainly consisting of *Pelteobagrus nitidus*, *Pseudobrama simoni*, *Carassius auratus,* and *Coilia nasus*. This shifted in summer to a dominance of *Coilia nasus* and *Hemibarbus maculatus*, with fish species primarily concentrated again in the 32–64 g size class. In autumn, fish species were mainly concentrated in the 64–128 g size class, predominantly comprising *Carassius auratus* and *Pseudobrama simoni*. By winter, the primary particle-size class shifted to 1024–2048 g, with dominant species including *Cyprinus carpio, Hypophthalmichthys molitrix,* and *Carassius auratus.*

During spring and summer, the catches consisted mainly of small fish, such as *Parabramis pekinensis* and *Coilia nasus*, resulting in high catch numbers and biomass, leading to a community structure dominated by smaller-sized fish. In autumn, the fish caught had increased in size due to fattening, reflected in a higher peak size compared to spring and summer. In winter, lower water temperatures forced small fish to winter in deeper areas, resulting in catches dominated by larger individuals.

The parameters of the standardized fish biomass particle-size spectrum reflect the structure and function of the ecosystem [36]. From 2022 to 2023, significant seasonal shifts and spatial differences were observed in the standardized biomass particle-size spectra. The curvature was highest in autumn and lowest in winter, indicating that the autumn spectrum line was relatively flat, while the winter line was steeper. Spatially, the estuarine waters had the highest curvature, while Gaoqing exhibited the lowest, suggesting that the estuary was least affected by external disturbances, while Gaoqing was most impacted.

Research has shown that fishing intensity affects the particle-size structure of fish, with the curvature of the standardized fish biomass particle-size spectrum decreasing as fishing intensity increases, leading to steeper spectra lines [37,38]. Overfishing results in a reduced proportion of larger fish within fish communities, an unstable particle-size structure, and a decrease in the average particle-size class, primarily featuring smaller sizes [39]. For instance, the peak particle-size classes for fish in Gaocun, Dong’a, and Gaoqing were all 16–32 g, while the estuarine area, being least disturbed, had a peak particle size in the 512–1024 g class.

The Shandong province area of the Yellow River is a complex ecosystem primarily influenced by human activities [40,41,42]. This study, which combined ABC curve analysis and biomass particle-size spectrum methods, investigated the degree of external disturbance and ecological characteristics of fish communities in the Shandong province area of the Yellow River: the fish community in the lower reaches were in a disturbed state, with an unstable community structure, a smaller range of particle-size classes, and changes in fish population structures. The biomass of larger particle-size classes decreased, while the biomass of fast-growing fish increased, leading to a community dominated by small-sized fish with rapid growth [43]. The curvature of the fish biomass particle-size spectra was small, and the curve was relatively steep, with only the estuarine area being least affected by external disturbances, while other survey areas experienced greater disturbance.

The study of the fish biomass particle-size spectra in the Shandong province area of the Yellow River provided valuable insights into the changes in fish community structure and enhances our understanding of the river basin’s ecological responses to natural disturbances and human activities. However, research on the fish biomass particle-size spectra in the Yellow River has been historically limited, resulting in a lack of comparable long-term data. This absence of comprehensive studies has hindered a thorough assessment of trends in particle-size structure and the ecological characteristics of fish communities in the region. To bridge this gap, it is essential to conduct continuous, long-term monitoring to systematically analyze variations in the fish particle-size distribution and community composition. A more robust dataset will allow for a deeper understanding of fish population dynamics and support the formulation of effective conservation and management strategies. In particular, adopting ecosystem-based fisheries management approaches—such as habitat restoration, sustainable fishing quotas, and stricter regulations—will be critical in ensuring the sustainable utilization and conservation of fishery resources in the Yellow River basin. By implementing these measures, we can help maintain ecological balance, preserve biodiversity, and secure the long-term viability of regional fisheries.

## 5. Conclusions

This study reveals the disturbed condition of the fish community in the Shandong province area of the Yellow River from 2022 to 2023. The dominance of small, fast-growing fish and notable changes in the fish biomass particle-size spectra signal underlying ecosystem imbalances. Ongoing monitoring and the adoption of ecosystem-based management strategies are crucial for ensuring the long-term health and sustainability of the river’s fish populations.

## Figures and Tables

**Figure 1 biology-14-00196-f001:**
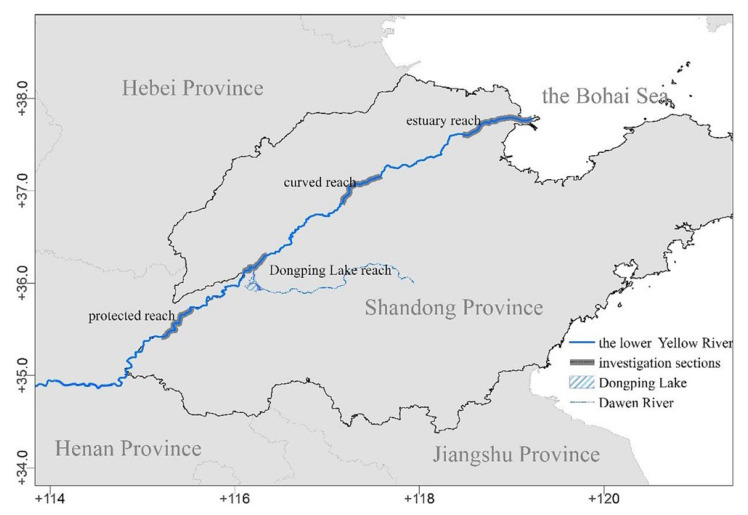
The sampling station locations in the lower Yellow River.

**Figure 2 biology-14-00196-f002:**
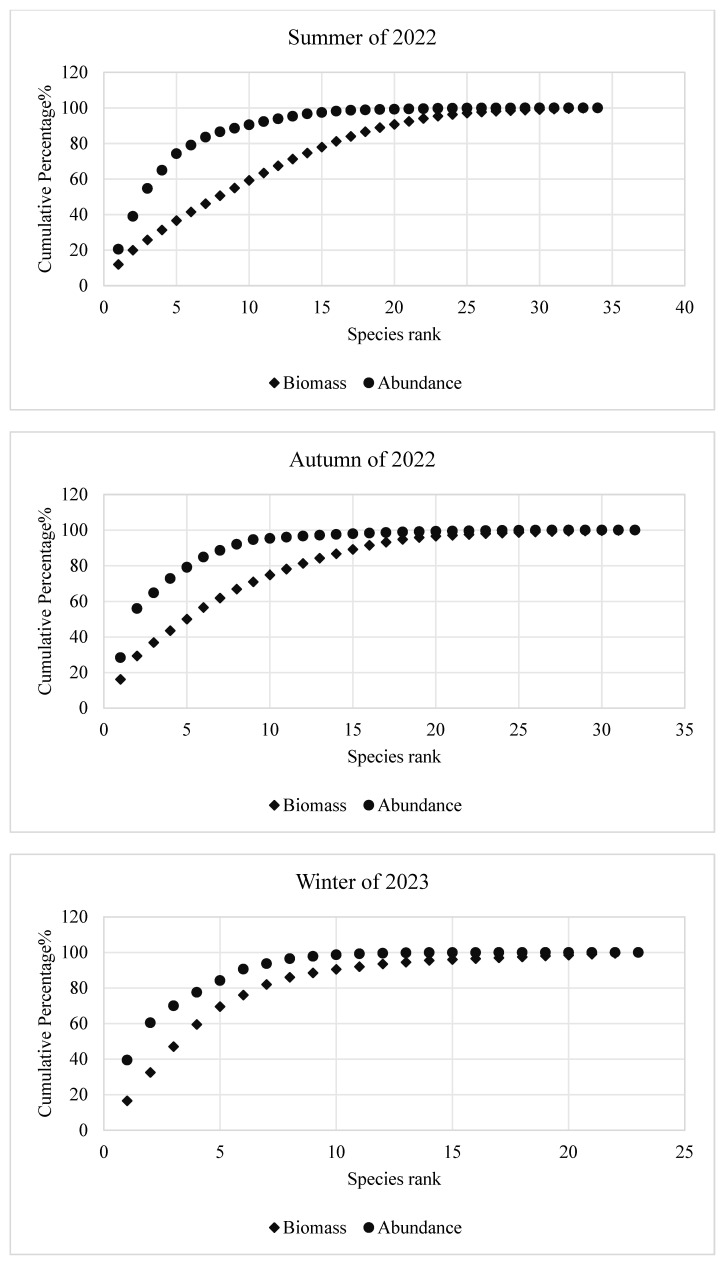
The ABC curves for the fish communities in the Shandong province area of the Yellow River.

**Figure 3 biology-14-00196-f003:**
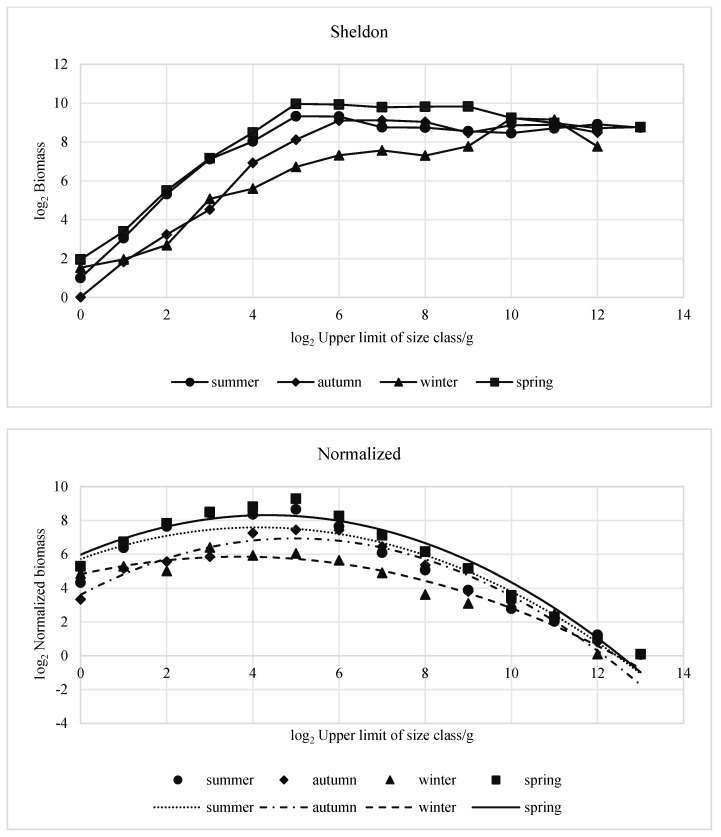
The seasonal variation of the fish biomass particle-size spectra in the Shandong province area of the Yellow River.

**Figure 4 biology-14-00196-f004:**
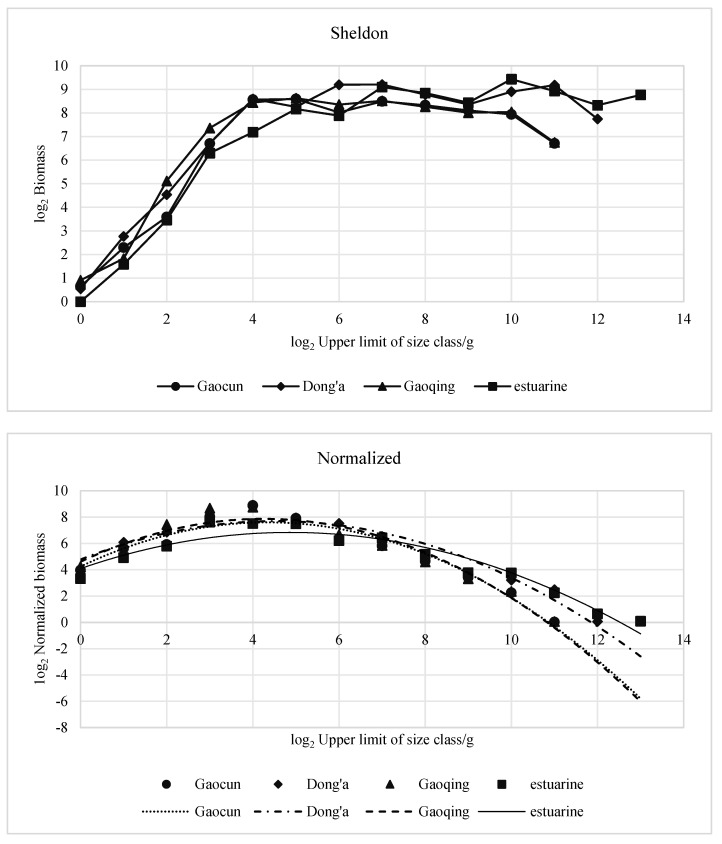
The regional variations in the fish biomass particle-size spectra in the Shandong province area of the Yellow River.

**Table 1 biology-14-00196-t001:** The composition of fish in the Shandong province area of the Yellow River.

Investigation Time	Summer of 2022	Autumn of 2022	Winter of 2023	Spring of 2023
Order	6	7	6	6
Family	13	11	7	12
Genus	32	28	22	31
Species	34	32	23	36
Catch rates in number/(ind.·net^−1^)	51.07	107.5	40.00	71.76
Catch rates in mass/(g·net^−1^)	16,547.67	9513.63	28,650.42	13,616.45
Average body mass/g	324.02	88.50	716.26	189.75
Dominant species	*Coilia nasus*, *Hemibarbus maculatus*, *Parabramis pekinensis*, *Saurogobio dabryi*	*Pseudobrama simon*, *Carassius auratus*, *Parabramis pekinensis*, *Hemiculter bleekeri*, *Squaliobarbus curriculus*	*Cyprinus carpio*, *Hypophthalmichthys molitrix*, *Carassius auratus*, *Squaliobarbus curriculus*	*Pelteobagrus nitidus*, *Pseudobrama simoni*, *Carassius auratus*, *Coilia nasus*, *Parabramis pekinensis*

**Table 2 biology-14-00196-t002:** Comparison of the main parameters of the fish biomass particle-size spectra.

Time	Size Range/g	Size Class of the Peak/g	Fitted Equation	R2
Summer	[0, 4096]	[32, 64]	y = −0.1096x^2^ + 0.905x + 5.7158	0.94
Autumn	[0, 2048]	[64, 128]	y = −0.1344x^2^ + 1.3367x + 3.6108	0.95
Winter	[0, 2048]	[1024, 2048]	y = −0.0762x^2^ + 0.5622x + 4.815	0.92
Spring	[0, 4096]	[32, 64]	y = −0.124x^2^ + 1.0773x + 5.9681	0.96

**Table 3 biology-14-00196-t003:** Interannual comparison of the normalized biomass particle-size spectra of fish in different regions.

Regional	Size Range/g	Size Class of the Peak/g	Fitted Equation	R2
Gaocun	[0, 1024]	[16, 32]	y = −0.1785x^2^ + 1.5491x + 4.2432	0.94
Dong’a	[0, 2048]	[16, 32]	y = −0.1493x^2^ + 1.3633x + 4.743	0.92
Gaoqing	[0, 1024]	[16, 32]	y = −0.1811x^2^ + 1.5374x + 4.6042	0.94
estuarine	[0, 4096]	[512, 1024]	y = −0.1163x^2^ + 1.1309x + 4.0868	0.93

## Data Availability

Data are contained within the article.

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
