# Peer review of "Temporal and Spatial Variation Characteristics of the Fish Biomass Particle-Size Spectra in the Shandong Province Area of the Yellow River"

_biology, 2025, doi:10.3390/biology14020196_

Round 1

Reviewer 1 Report

Comments and Suggestions for Authors

This manuscript has well studied the fish communities in the Shandong section of the Yellow River in China. It has significant contributions to protect the fish populations and to offer insights for sustainable management of Yellow River fish resources. I would recommend publication after minor revisions.

I’d recommend to change Table 1 into a figure to show the Shandong section of the Yellow River and the sampling sites. Make sure the locations listed in the study of e.g. Gaocun, Dong'a, and Gaoqing appear in the map.

I’d recommend to add an appendix or a supplementary to show the species list of the study.

The last figure should be “Figure 3”. It is better to use color lines in the figures.

Author Response

Point 1:I’d recommend to change Table 1 into a figure to show the Shandong section of the Yellow River and the sampling sites. Make sure the locations listed in the study of e.g. Gaocun, Dong'a, and Gaoqing appear in the map.

Response 1: I have revised the manuscript according to the reviewers' comments.

Point 2:I’d recommend to add an appendix or a supplementary to show the species list of the study.

Response 2: The supplementary tables have not been added. This article has already provided the names of dominant species, and the species list will be presented in another article.

Point 3:The last figure should be “Figure 3”. It is better to use color lines in the figures.

Response 3: I have revised the manuscript according to the reviewers' comments.

Reviewer 2 Report

Comments and Suggestions for Authors

In this study, the authors dynamically assessed the temporal and spatial changes in fish biomass in the Shandong province of the Yellow River over four seasons, revealing how overfishing affects fish populations and highlighting the importance of protecting regional biodiversity for food security and local livelihoods.

The authors found significant seasonal differences in the particle size spectrum of fish biomass in the Shandong Province section of the Yellow River. Relatively small-sized fish dominated the study area in spring and summer, slightly larger fish in autumn than in spring and summer, and relatively large-sized Cyprinus carpio and Hypophthalmichthys molitrix in winter. Regarding spatial variations, the authors found that larger-sized fish were distributed in the estuary areas, while smaller-sized species dominated the other sections.

The authors emphasized that human activities affect fish populations in the Yellow River in various ways. The most important of these is overfishing, which puts increasing pressure on large-sized species, especially due to their high economic value, and causes them to become extinct, and this selectivity shifts towards smaller and faster-growing species, causing critical changes in community structure and leading to a less diverse ecosystem. It was also emphasized that pollution and changes in habitat conditions could be factors that could disrupt ecosystem dynamics, leading to decreases in the overall biodiversity of fish populations in Shandong.

When the conclusions drawn from the findings are reached, the authors suggested that the effects of human activities on fish populations could be reduced and that effective management strategies are urgently needed to compensate for the fishery resources in the Yellow River and to promote their sustainability.

Considering all these, the authors should provide stronger suggestions for ecosystem-based fisheries management in order for their articles to contain resounding results. If this suggestion is implemented, the article could be suitable for publication.

Author Response

Point 1: The authors should provide stronger suggestions for ecosystem-based fisheries management in order for their articles to contain resounding results. If this suggestion is implemented, the article could be suitable for publication.

Response 1: I have revised the manuscript according to the reviewers' comments.

Reviewer 3 Report

Comments and Suggestions for Authors

The manuscript is clearly written and well-structured, making it easy to read and interpret. 

The sampling plan is well-conceived and clearly described. Covering four seasons in a year provides excellent temporal resolution to detect potential seasonal variations.

The applied methods are explained reasonably well and appear reproducible. However, in some cases, the interpretation of results is insufficiently detailed, as observed for the W value.

The conclusions are consistent with the findings presented in the analyses and graphs. The interpretation of the data appears quite accurate.

The cited bibliography is sufficiently up-to-date and relevant to the topics discussed. However, it should be reviewed, as in the row 359.

Some points should be corrected and revised. The following is a list of points to review.

·      In paragraph 2.2.2, the manuscript describes two possible classifications of particles but does not specify whether only one or both were used, nor the rationale for this choice. In line 153, it is stated that the classification of fish particles was conducted using the method of Sheldon et al., yet in paragraph 3.3, both classifications are discussed and illustrated. This discrepancy requires clarification.

·      In line 158, the calculation of average fish weight—obtained by dividing the total weight of all individuals by the number of specimens—does not account for size classes or developmental stages. This could result in inaccuracies when defining "small-sized" fish. For example, the presence of numerous juvenile specimens in certain seasons might introduce bias in the particle size classification. Were juvenile specimens excluded from the total catch before data analysis? If not, it could be a good idea to consider doing so.

·      The ABC curves support the hypothesis that the fish community in the river is heavily stressed by human activity throughout all seasons. However, the final part of paragraph 2.2.1 should be revised to provide a clearer explanation of the W values. While values are presented in line 197, the interpretation is unclear. Additionally, the "W%" shown in Figure 2 (likely representing weight) may cause confusion with the W value itself.

·      The figures and tables are sufficient in number and appropriate to support the scientific hypothesis. However, titles should be standardized (e.g., Figure 3), and more detailed captions should be provided, as suggested for Figure 3. For instance, identical names for lines and points in the figure make it challenging to interpret the graph without referring to the main text. In fig.2 W% in the ascissa (which I guess indicates weight) present in Figure 2 can be confused with the value W.

·      Check the sentence between lines 215 and 218 again. The lowest value of the coefficient is in winter.

Author Response

Point 1: In paragraph 2.2.2, the manuscript describes two possible classifications of particles but does not specify whether only one or both were used, nor the rationale for this choice. In line 153, it is stated that the classification of fish particles was conducted using the method of Sheldon et al., yet in paragraph 3.3, both classifications are discussed and illustrated. This discrepancy requires clarification.

Response 1: In Sections 3.3 and 3.4 of the results, both the Sheldon-type biomass particle size spectrum and the normalized biomass size spectrum (NBSS) were used for calculation and discussion. The classification of fish size classes in both methods is consistent and follows the approach proposed by Sheldon et al..

Point 2: In line 158, the calculation of average fish weight—obtained by dividing the total weight of all individuals by the number of specimens—does not account for size classes or developmental stages. This could result in inaccuracies when defining "small-sized" fish. For example, the presence of numerous juvenile specimens in certain seasons might introduce bias in the particle size classification. Were juvenile specimens excluded from the total catch before data analysis? If not, it could be a good idea to consider doing so.

Response 2: The average body weight is calculated by dividing the total weight of a specific fish species caught by the number of individuals. According to our sampling method, we did not exclude juvenile samples but included all individuals captured. The definition of small-sized fish is based on their maximum body length.

Point 3: The ABC curves support the hypothesis that the fish community in the river is heavily stressed by human activity throughout all seasons. However, the final part of paragraph 2.2.1 should be revised to provide a clearer explanation of the W values. While values are presented in line 197, the interpretation is unclear. Additionally, the "W%" shown in Figure 2 (likely representing weight) may cause confusion with the W value itself.

Response 3: Section 2.2.1 has already provided the calculation method and meaning of the W value. The annotation in Figure 2 has been revised.

Point 4: The figures and tables are sufficient in number and appropriate to support the scientific hypothesis. However, titles should be standardized (e.g., Figure 3), and more detailed captions should be provided, as suggested for Figure 3. For instance, identical names for lines and points in the figure make it challenging to interpret the graph without referring to the main text. In fig.2 W% in the ascissa (which I guess indicates weight) present in Figure 2 can be confused with the value W.

Response 4: The x-axis of Figure 2 has been modified; the chart titles of Figures 3 and 4 have been revised for differentiation.

Point 5: Check the sentence between lines 215 and 218 again. The lowest value of the coefficient is in winter.

Response 5: This part has been revised to: "the majority of particle size classes exhibited a gradual decreasing trend in individual numbers from spring to winter."